# An Ecological Sustainability Assessment Approach for Strategic Decision Making in International Shipping

**Xiaofang Wu [1,2,*] and Hsi-Chi Yang [1]**

[1] College of Harbour and Coastal Engineering, Jimei University, Xiamen 361021, China; hcyangse@gmail.com
[2] Fujian Shipping Research Institute, Jimei University, Xiamen 361021, China
*  Correspondence: xiaofang.wu@jmu.edu.cn

**Abstract:** Relatively little attention is currently paid to understanding the ecological impacts of international shipping, although ecological sustainability has become a necessary condition for developing international trade. Sustainability assessment, which identifies the sustainability-oriented effects of developmental activities for supporting decision-making, has been widely used. This study attempts to propose an ecological sustainability assessment approach to serve international shipping development based on the general assessment steps initiated by the OECD and the multi-dimensional decision making (MDDM) model. Compared with the existing sustainability assessment methods, the proposed approach is unlikely to be restricted to data acquisition, indicators evaluation, or causal recognition. Through a case study, the results recommend not only to prioritize avoiding the negative impacts of international shipping on noise, air, plants, water, and animals but also to promote continuous improvement of the local ecosystem and international shipping, particularly in the conditions of sediment and micro-organism communities of Xiamen. This proposed approach as a supplement to the current sustainability assessment methodology helps to make informative and integrative strategic sustainability decisions associated with international shipping.

**Keywords:** international shipping; ecological sustainability; assessment approach; strategic focus

## 1. Introduction

An ecosystem referred to as "*a dynamic complex of plant, animal and micro-organism communities and their non-living environment interacting as a functional unit*" [1], provides many goods and services such as food, energy, water, habitat, and recreation, which all support human society and economy. A healthy ecosystem is crucial for developing society. However, the ecosystem is becoming unhealthy due to the negative impacts of climate change, biodiversity loss, and pollution [2]. It was found that about 20 to 35 percent of mangrove ecosystems in coastal zones have been lost since 1980, making the coastal ecosystems vulnerable [2]. On such occasions, the World Commission on Environment and Development's 1987 report determined that "*the sustainability of ecosystems on which the global economy depends must be guaranteed*" [3]. Ecological sustainability has become "*the maintenance or restoration of the composition, structure, and processes of ecosystems over time and space*" [4]. This term hones in on conserving natural resources and biodiversity during industrialization [3,5] and has been regarded as the key approach toward sustainability for long-term reliance on ecological resources in development [6].

According to the sixth Global Environment Outlook, the environment and biodiversity of the oceans and coasts has been challenged by shipping due to its emissions, noise, discharge of sewage, ballast water, waste, and accidental releases [2]. It was estimated that the carbon dioxide emissions of worldwide transport would grow by 60% by 2050 when implementing current and announced policies [7] and the number of global premature mortality due to ships emissions (particulate matter, sulfur oxides, and nitrogen oxides) has increased from approximately 60,000 in 2002 to 250,000 in 2020 [8]. Confronted with these challenges, many academic works have concentrated on addressing sustainability aspects

in shipping networks [9], ship routing and scheduling [10,11], shipping operations [12], and berth allocation [13], by taking into consideration of fuel-saving and emission reduction policies or strategies [10]. However, the impacts of such strategies seem to be geographically diverse and variable over time. Some literature suggested understanding the status of international shipping and regional ecosystems and the interactions before introducing sustainable actions to shipping stakeholders [14]. The Organization for Economic Co-operation and Development (OECD) advised developing a sustainability assessment approach for understanding the sustainability status adequately, supporting regional or local strategies [15]. On the other hand, many governments, non-profit organizations, companies, and other related agencies also became aware of sustainability-oriented solutions for international shipping to take responsibility, while most efforts kept their eyes on piecemeal and passive improvements of ecological performance [16,17]. The situation is detrimental to work toward the long-term blueprint of sustainability in complex human and natural systems, not least because of the lack of appropriate approaches or processes with a sufficient understanding of sustainability impacts related to international shipping to support a systematic, active, and strategic decision-making [18].

Currently, a sustainability assessment tool is commonly used to understand the impacts of industrial activities and to seek ecological, economic, and social sustainability measures for supporting decision-making [15,19]. This tool contributes to integrating sustainability issues into the decision-making process and bringing about future sustainable policies, plans, and projects [20]. In 2008, the OECD organized a workshop to discuss sustainability assessment methodologies and practices and proposed a guideline for the basic elements, processes, and nature of sustainability impact assessment, published in 2010 [15,21]. The assessment steps thereof include relevance analysis (screening and scoping), delineation (selecting tools or methodologies), impact analysis, and optimization (proposing measures and results) [21].

By searching the literature with keywords such as "sustainability assessment" in the Web of Science and by reading their abstracts, numerous evaluation methods can also be identified. For instance, a multi-criteria decision making (MCDM) method was adopted to generate sustainability alternatives by selecting indicators, by assigning weights, and by evaluating and ranking alternatives [22,23]. With a similar purpose, another method, the multi-dimensional decision making (MDDM) approach, considered all available data and included expert' participants to help in decision making [22,24]. Moreover, several researchers used life cycle assessments (LCAs) and life cycle sustainability assessments (LCSAs) to evaluate the environmental effects in a life-cycle view [25,26]. Their input–output approaches have also been applied for other methods such as carbon/ecological footprint, input–output analysis, and data envelopment analysis (DEA) to measure carbon dioxide emissions or ecological impacts [27–30]. Additionally, there is another method of DPSIR (driver pressure state impact response), based on the recognition of cause effect relationships, that tries to model the drivers, pressures, states, impacts, and responses of sustainability [31]. Furthermore, the sustainability impacts were quantified to connect the ecological and economic dimensions using emergy analysis and economic value added (EVA) [32,33] and the uncertainty of assessment was analyzed through evidential reasoning (ER) [34].

However, based on the literature search with keywords such as "shipping" and "sustainability assessment" in the Scopus database, only 12 papers were retrieved till 7 October 2021. After reading the literature, we can just find a few studies that used a tool of MCDM to support the sustainability assessment of abatement technologies and marine fuels [35,36]. Such an absence has weighed on understanding the sustainability impacts of international shipping activities as well as on identifying their potential effects or detecting the trade-offs among different sectors [15]. The limited focus of the current literature may lead to a limitation of findings, so as to affect the effectiveness of decisions. Thus, how sustainability assessment tools support international shipping's strategic decision-making is questioned. Given that ecological sustainability lays the foundation for sustainability, this

study aims to develop a sustainability assessment approach from ecological perspectives for carrying out strategic decision making within the domain of international shipping, after discussing the existing sustainability assessment tools. Then, we test the constructed approach through a case study and conclude the research.

## 2. Materials and Methods

An ecological sustainability assessment in international shipping is a process that integrates the goal of ecological sustainability into the assessment of international shipping activities to support strategic decision-making. Rather than defining this phrase beyond the objective of this study, we concentrate on the construction of an assessment approach. This study falls back to the leading and widely used procedure provided by OECD [21] concerning existing specific evaluation methods, e.g., MCDM, MDDM, LCA/LCSA, carbon/ecological footprint, input–output analysis, DEA, DPSIR model, emergy analysis, EVA, and ER.

However, international shipping, which involves multiple disciplines, crosses regions and has multiple stakeholders, faces many difficulties in the accessibility of ecological data and the recognition of cause effect relationships [17]. This may hinder the utilization of LCA/LCSA [37], carbon/ecological footprint [38], input–output analysis [39], and DEA [30], which rely considerably on sufficient data as well as on the DPSIR model [31] for its restrictions on causal recognition. The methods of emergy analysis, EVA, and ER [32–34] are expected to be used for further analysis of economic valuation or uncertainty of impacts. Both MCDM and MDDM aim to provide alternatives based on indicator evaluations and experts' judgment, while the MDDM may better fit strategic decision making in the absence of sufficient data, for a combination of a top-down strategic ecological sustainability focus and a bottom-up broader data collection [14,24,40]. Combining experts' opinions and bottom-up information has also shown advantages in reducing the uncertainty associated with expert judgment [24]. Therefore, this study adopts the MDDM method to help analyze ecological sustainability impacts of international shipping and where an evaluation model can be applied [24].

Due to a lack of accessible practices on ecological sustainability assessment of international shipping, a case study may be an attractive means to test and verify the proposed approach. Combining documentary analysis and expert participation [41], the approach allows for a better mutual understanding of the local ecosystem and international shipping activities and assists in the exploration of the needs to sustain the development of international shipping from an ecological perspective. Materials that support the assessment include papers, books, plans, yearbooks, bulletins, and reports from literature databases and related official websites.

## 3. Proposed Approach

Based on the OECD's general steps of sustainability impact assessment [21] and the MDDM method [14], this study proposes four steps in ecological sustainability assessment in international shipping, as follows.

Step 1: Scoping.

The initial step is to determine the extent and depth of the assessment of international shipping. Under globalization, technological advances, and operational integration in the modern era, as the lifeblood of the global economy, international shipping that serves "door-to-door" intermodal transportation services integrates maritime transportation, port operations, and land transportation for "door-to-door" shipping services [16,17]. This study does not commit to defining modern international shipping but identifies their activities that matter regarding ecological sustainability issues, such as multi-modal transportation, storage, loading, unloading, handling, packaging, and distribution processing. Given the components of the ecosystem [21], the indicators that matter regarding interactions between ecosystems and international shipping activities are displayed in Table 1. This study concerns their primary ecological effects rather than secondary damages such as property.

**Table 1.** Indicators for ecological sustainability assessment in international shipping.

| Categories | Indicators | |
|---|---|---|
| Plant | Plant community, landscape, ecologically sensitive areas | |
| Animal | Habitat, protected or endangered species, ecologically sensitive areas, animal health | |
| Micro-organism communities | Diversity, flora, fauna, protected or endangered species, ecologically sensitive areas | |
| Non-living environment | Air | Air pollutants (e.g., $SO_X$ (sulfur oxide), $NO_X$ (nitrogen oxide), CO (carbon monoxide), $PM_{2.5}$ (particulate matter, diameter of which $\leq$ 2.5 microns), $PM_{10}$ (particulate matter, diameter of which $\leq$ 10 microns), HC (hydrocarbons)); greenhouse gases (e.g., $CO_2$ (carbon dioxide), $CH_4$ (methane)); ODS (ozone depleting substances); PBTs (persistent bioaccumulative and toxic substances) |
| | Water | Concentrations of elements in water quality, such as nitrogen, sulfide, floating substance, etc. |
| | Waste | Waste production, treatment, disposal of, and recycling |
| | Sediment | Concentrations of elements in sediments such as organic carbon, heavy metals, polychlorinated biphenyls, etc. |
| | Noise | Equivalent sound level |
| | Soil | Concentrations of elements in soil environmental quality such as mercury and lead, soil erosion |

Step 2: Data collection.

On a basis of the MDDM model, there is a need to collect as much as available data in association with the quality of local ecosystems and the status of international shipping activities as a reference for supporting the assessment, especially for expert judgment. Combing the bottom-up information and experts' knowledge contributes to generating effective and reliable decisions [24]. This study suggests collecting plans, yearbooks, bulletins, and reports from official websites to help understand ecological conditions. For another, papers, books, and reports found in literature databases and related official websites may benefit the understanding of international shipping for proactive cross-department exploration. The ecological indicators offered by Table 1 can also provide a reference.

Step 3: Assessment.

Following the MDDM method, the ecological sustainability impact analysis in international shipping must be separated into two parts. The first is to understand the conditions of international shipping and ecosystems. Based on previous information, a score of 1, 2, or 3, indicating bad, middle, and good, respectively, would be provided by experts [24]. Experts selection criteria include: (1) expertise in international shipping operations and environmental management; (2) approval of sustainable or green development; (3) being representative in understanding the regional or local industrial development and ecological status. Second, a model of I, C, R (Impact, Confidence, Relationship) would also be scored to express the ecological impacts caused by international shipping, i.e., scores for I would be from −3, −2, −1, 0, 1, 2, 3 to indicate positive or negative impact; C would be a value

between 0 and 1 to express the scorers' confidence; and R would be 0, 1, 2, or 3, to illustrate their relationships [24]. Finally, the impacts are valued with the expression as follows [24]:

$$S_{ki}^j = I_{ki}^j \times C_{ki}^j \times R_{ki}^j \ (k = 1, 2, \ldots, q; \ i = 1, 2, \ldots, m; \ j = 1, 2, \ldots, n) \tag{1}$$

$$S_i^j = \frac{\sum_k^q I_{ki}^j \times C_{ki}^j \times R_{ki}^j}{q} \ (k = 1, 2, \ldots, q; \ i = 1, 2, \ldots, m; \ j = 1, 2, \ldots, n) \tag{2}$$

where $k$ presents the expert's serial number, $q$ indicates the total number of experts, $i$ is the international shipping activity, $m$ means the total number of international shipping activities, $j$ is the ecological element, $n$ shows the total number of ecological elements, $S_{ki}^j$ represents the expert $k'$ s opinion on the impacts of shipping activity $i$ on ecological element $j$, and $S_i^j$ represents the impacts of shipping activity $i$ on ecological element $j$ [14].

Step 4: Recommendation.

Focusing on the assessment, this study attempts not only to avoid considerable negative ecological impacts of international shipping but also to improve ecological conditions and international shipping activities regarding further outcomes. Through combining a broad data collection and opinions from expert participants, decisions would be informative and effective.

## 4. Case Study and Results

After providing a conceptual and methodological approach, we implement and test a pilot application. In this study, we chose the international shipping port in Xiamen, which is located in southeast China, and which faces the Taiwan Strait. Being a backbone of the national transport network, Xiamen has an international hub seaport and became an international comprehensive transportation hub [42]. The ecological sustainability assessment of international shipping in Xiamen is beneficial for supporting governments or enterprises interested in creating sustainable policies, plans, or programs.

### 4.1. Scoping

Based on practical evidence, international shipping activities in the Xiamen area include rail or road transportation, port operations, and maritime transportation. Among them, port operations may consist of activities such as loading, unloading, handling, storage, and packaging. Ecological sustainability impacts caused by possible international shipping activities focus on the direct effects on plants, animals, micro-organism communities, and non-living environments, as Table 1 shows.

### 4.2. Data Collection

This study searched the Scopus database using the keywords "maritime OR logistics OR shipping OR port OR harbor AND Xiamen" to cover a broad scope of information. This combination of keywords reflects the maritime focus. A sum of 189 published papers was collected for a period dating up until 17 July 2021. We then excluded the literature irrelevant to international shipping activities and ecological issues in Xiamen based on their titles and abstracts. As a result, 48 papers remained for further analysis. By searching official websites, including from the Xiamen Statistics Bureau, Xiamen Port Authority, Xiamen Municipal Ecological Environment Bureau, Xiamen Municipal Natural Resources and Planning Bureau, Xiamen Municipal Bureau of Ocean Development, materials that embrace the Yearbooks of Xiamen Special Economic Zone, and bulletins of Xiamen environmental quality. Eventually, a description of the ecosystems and international shipping conditions in Xiamen was compiled and is shown in Tables A1 and A2. Meanwhile, we also compared the ecological status of Xiamen to one of the top 10 greenest cities in the world, Singapore [43], and drew comparisons on the conditions of international shipping between Xiamen and other advanced areas as reference.

*4.3. Assessment*

By adopting the MDDM model, the assessment for impacts on the ecological environment was undertaken by experts from universities, governments, research institutes, and companies. A broad scope of information is provided in Tables A1 and A2, which will help experts effectively judge the status of local ecosystems and international shipping, and their impacts, and will help them rigorously decide on strategies toward sustainability. In this case, fifteen experts who have impressive professional knowledge on international shipping operations and environmental management in Xiamen were invited to perform the assessment while eight experts eventually took part in this program as a result of creating a high confidence score of 0.5, including Expert A from a university; Experts B and C from the government; Experts D and E from research institutes; and Experts F, G, and H from companies.

4.3.1. Assessing the Conditions of the Ecosystem and International Shipping

Through the experts' participation, scores of 1, 2, and 3, which respectively show poor, medium, and good conditions, were given to indicate the status of Xiamen ecosystems and international shipping. Figures 1 and 2 demonstrate the results. Given the figures, few scores have two-point gaps, and the average scores were close to the median. There is also consistency among the scores. On average, we can find that plants, animals, and soil are in good status, relatively, while the conditions of sediment and micro-organism communities are slightly poor. For the other study, there appear to be noticeable negative impacts on the water by maritime transportation, on the noise and air by port operations, and on the air by rail or road transportation.

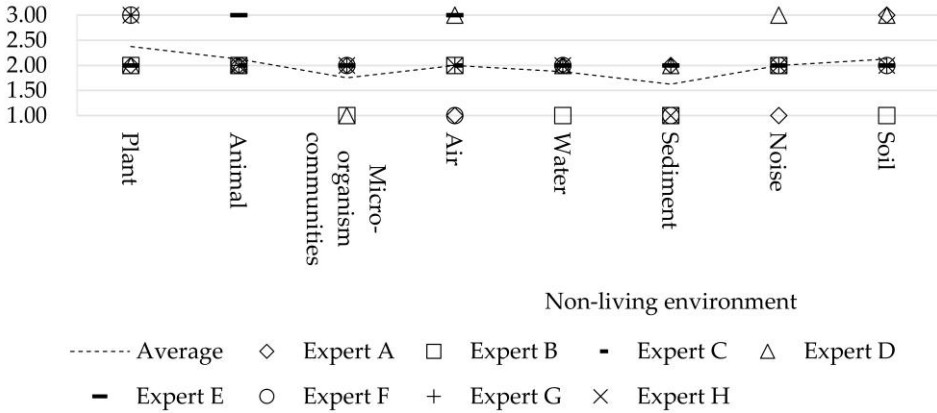

**Figure 1.** The assessment of ecosystem conditions in Xiamen.

4.3.2. Assessing the Ecological Impacts of International Shipping

With the local ecosystem and international shipping conditions in mind, the experts provide their opinions on the ecological impacts of international shipping by scoring the I, C, R of the MDDM model and by calculating them using Equation (1) and Equation (2). Figure 3 presents the results where opinions from experts ($S_{ik}^{j}$) are indicated by different kinds of scattered geometric shapes while final views ($S_{i}^{j}$) are connected by a dotted line. It is, however, not visible that most of the experts scored the impacts ranging from $-3$ to $0$, with the average scores being on a scale of $-0.05$ to $-4.65$. Through expert consultation, the factors that are scored, on average, less than $-2.00$ should be given special consideration in respect to the avoidance of negative impacts. As a result, it can be found that maritime transportation has a noticeable negative influence on animals; port operations affect water and noise negatively; and rail or road transportation causes significant negative effects on noise, air, and plants. The local ecosystems, in contrast, are influenced more by land transportation.

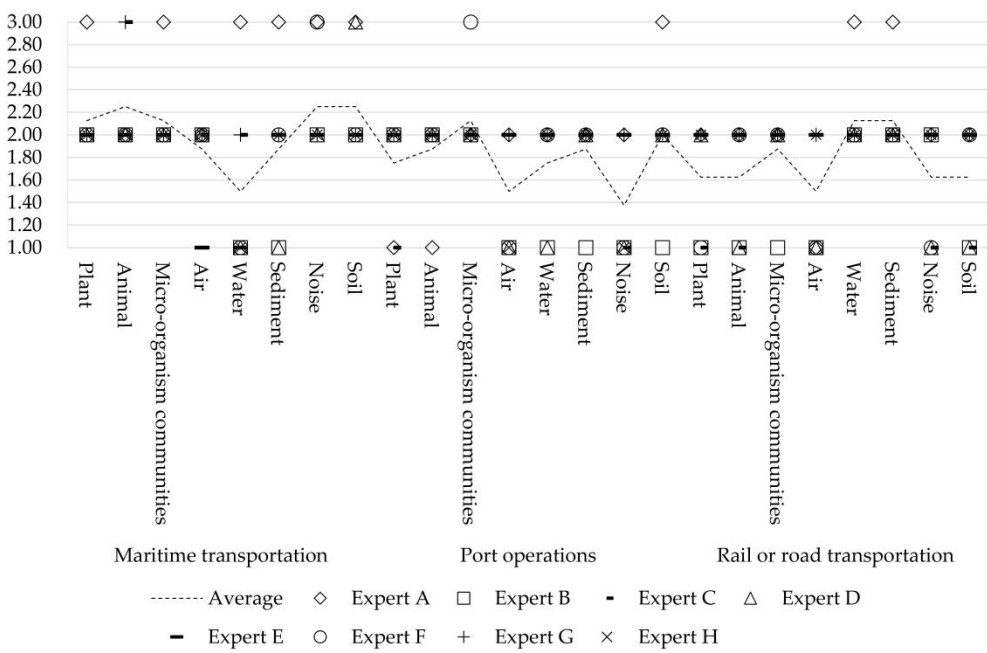

**Figure 2.** The assessment of international shipping conditions in Xiamen.

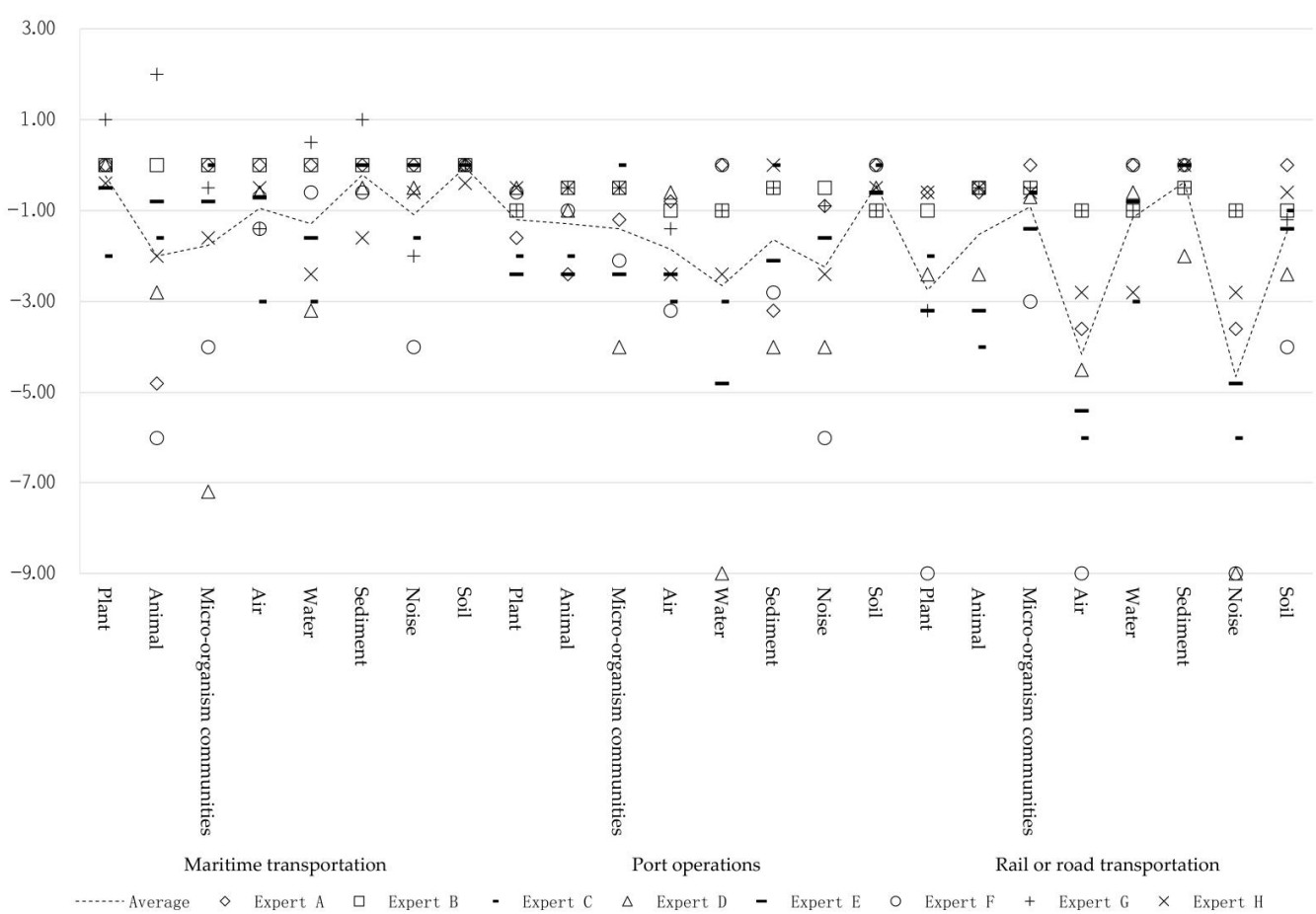

**Figure 3.** The assessment of the ecological impacts of international shipping in Xiamen.

*4.4. Recommendation*

This study provides strategic options to avoid the considerable negative effects of international shipping based on the assessment results. Due to the significant negative ecological impacts from land transportation, it is recommended to stimulate intermodal transport in Xiamen, moving freight off road to a certain extent. Other recommendations include:

- the first is to concentrate on the need to reduce noise and protect air and plants from land transportation activities in terms of their maximum absolute values. Effects of traffic noise could probably be reduced by using quieter engines or tires or constructing noise barriers or insulation; installation of exhaust after-treatment devices can also be helpful to reduce air pollution from road transportation, and building barriers might be able to reduce vegetation exposure to air pollution.
- the second is to alter port operations, which have considerable negative effects on water and noise. Based on a previous study of [44], there is a risk of oil spills from vessels threatening the water quality of Xiamen port areas. Therefore, we may need to strengthen risk prevention. Furthermore, it suggests improving the traffic efficiency at the port entrance according to the current situation in Xiamen [44].
- third, maritime transportation operations should keep a watchful eye on animal habitats. As a coastal city of Xiamen, there are some sensitive areas near the port areas such as protected zones for Chinese white dolphins and mangroves [44]. Conflicts between shipping activities and ecological issues need concerns.

Further improvements of local ecological conditions and international shipping operations are recommended to combat the conditions of sediment and micro-organism communities in the Xiamen area. This includes protecting from noise from port operations, reducing air emissions caused by port operations and land transportation, and altering the effects of maritime transportation on water. On such occasions, several cost-effective ecosystem-based mitigation measures would be preferable for nature conservation, such as increasing green spaces to mitigate the effects of port operations, using renewable energy to reduce emission sources, or constructing natural or green infrastructure to restore natural ecosystems [45–47].

## 5. Discussion

Compared with the current sustainability assessment approach, this ecological sustainability assessment approach has the following advantages: (1) gathering as much available information, instead of relying on indicators, to help reach a satisfactory assessment; (2) creating a data-driven assessment, rather than basing it on the recognition of cause effect relationships that are subject to uncertain and complex conditions of international shipping; (3) combining the expert opinions and bottom-up information to promote the reliability and effectiveness of strategic decision-making; and (4) not only targeting an avoidance of the considerable negative effects of international shipping activities but also suggesting to further improve the conditions of the local ecosystem and maritime activities because of potential risks and "win–win" considerations.

Through a case application, the results show not only the pressing negative impacts caused by international shipping activities but also the need for improvement on the local ecosystem and international shipping. The two-part considerations enable the decision makers to take actions step-by-step to avoid or mitigate the effects of maritime activities and to improve ecological conditions. Generally, the measures to avoid damage may be directed to passively solving the problems, while the actions improving the conditions may be more of concern regarding potential effects and resilience from systematical and long-term views.

In recent decades, an emerging concept of nature-based solutions not only caught the attention of researchers [45] but also was initiated by the International Union for Conservation of Nature (IUCN) and the European Commission to help us better use, manage, restore, or create ecosystems by learning from nature, complementary to existing engineering or technological solutions [46]. Studies estimate that the implementation of

nature-based solutions can offer about 30% of the cost-effective mitigation required by 2030 to stabilize climate warming to below 2 °C [47]. We find them to be better options of improvement to the conditions of the local ecosystem toward sustainability for conserving nature with regard to international shipping, such as prioritizing the use of renewable energy and the construction of ecological buffers, instead of installing noise reducers or scrubbers.

Our study of an ecological sustainability assessment approach is uncovering a way in which a strategic decision-making system can be facilitated by combing through a broad scope of information as well as expert knowledge. This study provides an ecology-based sustainability assessment approach as a supplement to the current sustainability assessment methodology to serve strategic decision-making associated with international shipping, although limitations exist in the assessment process, e.g., the bias of data acquisition and the restriction of causality identification confined by resources and funds. Nevertheless, a more detailed evaluation for the indicators in Table 1 would be beneficial in understanding the complex and uncertain business-ecology interactions and more factual evidence may help improve the credibility and effectiveness of the assessment, complementary to the literature analysis. Concerning the three aspects of sustainability, i.e., economy, society, and ecology [15,19], trade-offs between the three are also needed. When time and funds are available, a system dynamic approach that studies the interrelationships among several relevant variables of industrial behavior [48] may contribute to understanding and measuring the causal impacts of shipping activities on ecosystems for a better basis of decision making because ecological sustainability is a dynamic task in essence.

## 6. Conclusions

Ecological sustainability has been of survival value to the human race. As the backbone of international trade, international shipping, which integrates maritime transportation, port operations, and land transportation to serve door-to-door shipping, aims to sustain its activities but its efforts are fragmented and passive. This situation may hinder strategic and holistic thinking, collaborative and cooperative efforts, and cost-efficient and effective plans. In recent decades, a sustainability assessment tool has been commonly used to help make systematic and rigorous decisions toward sustainability. There are also several evaluation methods, such as MCDM, MDDM, LCA, input–output analysis, DPSIR model, EVA, and ER, that are used to assist the sustainability assessment process, although most of them are still restricted to data acquisition, indicator evaluations, and causal identification.

Under this circumstance, this study adopts the leading sustainability impacts assessment procedure and the MDDM method to construct an ecological sustainability assessment approach, which combines a top-down strategic ecological sustainability focus and a bottom-up information assessment. It complements the existing sustainability assessment methodology to fit the strategic decision-making related to international shipping under the current situations of literature. Through the application of an information evaluation and through expert participation, this assessment approach can be used in the absence of sufficient data and causal recognition. It also contributes to not only identifying the ecological effects of international shipping but also recognizing the need to improve the local ecosystem with regard to international shipping. Through a case study, the results indicated the significant negative effects of land transportation on noise, air, and plants; the considerable impacts of port operations on water and noise; and the effects on animals by maritime transportation. Furthermore, improvements on sediment and micro-organism community conditions in Xiamen as well as the impacts of international shipping on noise, air, and water should be paid attention to. As alternatives, nature-based solutions that learn from nature are conducive to achieving sustainability, especially for continuous improvements.

Despite the limitations on data accessibility and expert participation, the ecological sustainability assessment approach is necessary to systematically and strategically understand the status of international shipping and local ecosystems for supporting strategic

decision making, versus passively solving specific problems in pieces. In the future, more detailed and dynamic evaluations such as using a system dynamic approach to understand the causality of interactions between shipping activities and ecosystems might be beneficial to further data collection and analysis, and wider applications of the proposed assessment approach are good choices to improve the reliability and effectiveness of strategic decision making, if time and funds permit. We also suggest building alliances and relationships to gain support for the assessment as well as policies or strategy making, especially in the context of "big data". To handle the sustainability issue, further research should enable trade-offs of the ecological, economic, and social impacts of international shipping.

**Author Contributions:** X.W. designed and drafted the manuscript. H.-C.Y. reviewed and commented. All authors have read and agreed to the published version of the manuscript.

**Funding:** This research was funded by the Cultivation Fund for High-level Research of Transportation Engineering Discipline of Jimei University, grant number HHXY2020017, the Natural Science Foundation of Fujian Province, China, grant number 2020J05143, and the Jimei University Scientific Research Starting Foundation, grant number ZQ2019037.

**Institutional Review Board Statement:** Not applicable.

**Informed Consent Statement:** Not applicable.

**Acknowledgments:** We thank experts who have participated in and supported the assessment. We also thank Jianwu Cai for comments on the earlier version of the manuscript.

**Conflicts of Interest:** The authors declare no conflict of interest.

# Appendix A

**Table A1.** The ecological conditions of Xiamen and their comparison to Singapore.

| Indicators | | A Description of Conditions in Xiamen | A Description of Conditions in Singapore |
|---|---|---|---|
| Plant | | Until the year 2019, the ratios of green area and green coverage in Xiamen City were 40.9% and 45.1%, respectively, and the per capita green area of the park was 15.6 $m^2$ [49]. The natural conditions of Xiamen Bay are suitable for mangrove habitats and the area of mangroves accounted for about 0.04% of the Xiamen's area in 2017 [44,50]. | In 2019, the ratio of greenery was more than 40% in Singapore, and the per capita green area of the park was 7.9 $m^2$ [51]. The area of mangroves accounted for about 0.1% of Singapore's area in 2012 [52]. |
| Animal | | From 2011 to 2014, the distribution of the humpback dolphins in Xiamen shifted from inner harbors to peripheral waters, and their group density was significantly decreased in marine protected areas and conservation zones [53]. In 2019, the number of Chinese white dolphins maintained a steady population, and the Chestnut-throated Bee-eater showed slight growth in numbers [54]. | No apparent changes can be found in the fish species composition and loss in the open water of Singapore's mainland, except for a decline in abundance in the 1970s [55]. |
| Micro-organism communities | | Between 1980 and 1990, the benthic macroinvertebrates biodiversity in Xiamen bay remained high but decreased rapidly from 2005 to 2007 [56]. From 2013 to 2016, the abundances of phytoplankton, zooplankton, and benthos had been boosted slightly, while their species richness slightly declined [44,56]. The altered environmental conditions, eutrophication, and exotic species intrusion influenced the structure of the plankton community and red tide occurred frequently in Xiamen [57]. | Due to landfill dumping, the abundance of the benthic community and the familial diversity decreased in the 1990s near the offshore island in the Singapore Strait, and biological communities were threatened by reclamation for port extension in the 1970s [55]. |
| Non-living environment | Air | Xiamen has a subtropical maritime climate [58]. The $PM_{2.5}$ and $PM_{10}$ were alleviated in Xiamen harbor during the years 2014 to 2018, but their average concentration exceeded the National Ambient Air Quality Standards in winter and spring [59]. In 2019, the air quality of Xiamen was at 97.5% on the excellent and good air quality index, down by 1.1% compared with that in 2018, and is ranked the fourth among 168 cities in China, down from second position in 2018 [54]. Nine days were marked as slightly polluted, caused by $PM_{2.5}$ and ozone in 2019 [44]. Although $NO_2$ was not included, the number of slight pollution days showed an increase compared with that in 2018 [54]. The annual average concentrations of $SO_2$, $NO_2$, and CO can meet the first-level national standard as well as the concentration of PM10, which was at the second level in 2018 [54]. The frequency of acid rain occurrence was 60%, down from that in 2018 [54]. | In 2019, the air quality of Singapore was at 97% on the good and moderate air pollutant standards index, down by 3% compared with that in 2018 [60,61]. In that year, the average $PM_{2.5}$ concentration of 19 $\mu g/m^3$ exceeded the World Health Organization's recommended target of 10 $\mu g/m^3$ [62]. The air pollution levels of Singapore ranked the 52nd worst of the 98 countries based on $PM_{2.5}$ [62]. |

**Table A1.** *Cont.*

| Indicators | | A Description of Conditions in Xiamen | A Description of Conditions in Singapore |
|---|---|---|---|
| | Water | Xiamen has regular half-day tides [58]. In 2017, the areas that complied with the Class I and Class II criteria in Sea Water Quality Standard (GB 3097-1997) in China accounted for 69.1% of the total area of Xiamen Bay, up by 3.3% compared with that in 2016 [63]. In 2019, the water quality at seven of the eleven offshore water quality monitoring sites met the requirements of the Marine Environmental Function Zoning [54]. The main pollutants were all active phosphate and inorganic nitrogen, and both dropped in annual average concentrations in 2019, compared with those in 2018 [54]. Eutrophication ranging from mild to severe happened in the seawater around Xiamen Island [63]. In coastal areas, the abundance of microplastics in surface seawater ranged from 103 particles/m$^3$ to 2017 particles/m$^3$ [64]. In surface seawater, concentrations of dissolved polycyclic aromatic hydrocarbons varied from 18.1 ng/L to 248 ng/L [64]. | The water quality of Singapore's popular recreational beaches was graded as "good" in recent years, based on the World Health Organization's guidelines [65]. |
| | Sediment | As a deep water harbor, the average sediment concentration in the bayou is 0.04–0.06 kg/m$^3$, and the tidal volume is about 700 million m$^3$ in Xiamen [66]. In 2017, there was no significant change in the environmental quality of sediments in Xiamen's offshore waters; the concentrations of SO$_X$ and certain heavy metals such as cuprum and plumbum in surface sediments exceeded the first level in the Sea Sediment Quality Standard (GB 18668-2002) in China [63]. In sediment, the site near the Xiamen Port exhibited a high concentration of perfluoroalkyl and polyfluoroalkyl substances [67]. The abundance of micro-plastics was 76 particles/kg to 333 particles/kg in coastal sediments [64]. | Over the past decades, land reclamation and dredging increased the sedimentation rate in Singapore, with rates of 10 to 90 mg/cm$^2$/day around the southern island [68]. In the 2000s, the suspended sediment concentration along the west coast of the southern island was about 0.005–0.02 kg/m$^3$ [68]. |
| | Noise | The daytime road traffic noise received an average equivalent sound level of 67.2 dB(A), less than the excellent level limits of 68.0 dB(A), but showed an augmentation of 0.8 dB(A) to that in 2018 [54]. | Due to modifying vertical exhaust, overloading, and speeding, the road traffic noise emission was excessive in Singapore [69]. |
| | Soil | Excluding medical waste, the disposal rate of hazardous waste was 99.8% in 2020, down by 0.2% compared with that in 2019 [54]. | In 2019, the overall rate of recycling in Singapore decreased to 60% compared with 80% in 2018 [60,61]. |

**Table A2.** The conditions of international shipping in Xiamen and their comparison to other cases.

| Activities | Indicators | | A Description of Conditions of Xiamen International Shipping | A Description of Conditions of Other Cases |
|---|---|---|---|---|
| Maritime transportation | Animal | | In contrast with the impacts of distribution patterns of the humpback dolphin community caused by coastal tourism and constructions, shipping activities are not the primary cause [54]. | In the 2000s, the Indo-Pacific Humpbacked dolphin often haunted the port areas of Singapore, but other marine mammals were rare [55]. |
| | Non-living environment | Air | The ship emissions of $SO_2$, $NO_X$, HC, CO, $PM_{2.5}$, and $PM_{10}$ in the Ship Emission Control Area in Xiamen in 2018 were about 2816, 10544, 706, 1755, 491, and 592 tons, respectively [70]. The PM emission in Xiamen presents less than that from ships in Dalian Port [70]. The shipping activities were likely sources of refractory black carbon aerosol in Xiamen [71]. | Traffic emissions are the main sources of air pollution in Singapore [62]. The ship emissions of $SO_X$, $NO_X$, $CH_4$, CO, PM, and $CO_2$ in the Ship Emission Control Area in Singapore in 2018 were about 41300, 32200, 4.9, 2760, 3740, and 2720000 tons, respectively [72]. |
| | | Water | Contributing to the shipping activities, the concentrations of Benzoapyrene in the seawater from the Western Xiamen Harbor were high and brought risk to the fish eggs [73]. In 2015, the concentrations of polybrominated diphenyl ethers, polycyclic aromatic hydrocarbons, and organochlorine pesticides as well as the characteristics of water-soluble inorganic ions of $PM_{2.5}$ and aerosol acidity, increased due to shipping activities [74,75]. Risks of oil spills and ballast water discharging also exist [44]. | The waters surrounding the Port of Singapore are generally turbid and eutrophic [76]. Risks of ballast water discharging existed for species invasion, and the shipping industry may be in charge of half to two-thirds of the bio-invasions in Singapore [76,77]. In 2003, the polychlorinated biphenyl concentrations increased greatly due to shipping activities [78]. |
| Port operations | Non-living environment | Air | The mean concentration of volatile organic compounds in the Xiamen Port area was higher than that in the residential zone [79]. In the summer of 2015, the daily $PM_{2.5}$ concentrations in the port area exceeded the Chinese Ambient Air Quality Standards and the ship emissions increased the mass concentration of metals in $PM_{2.5}$ [80]. | The emission external cost per thousand throughputs in Xiamen Port was higher than that of Shanghai Yangshan port in 2009 [81,82]. In 2014, the emission intensity of $SO_X$ and carbon dioxide per throughput in Xiamen Port was higher than those in the port of Long Beach [44]. |
| | | Water | From 2010 to 2015, the sea areas for port use increased from 16.50 to 17.56 $km^2$ but decreased from 20.06 $km^2$ to 9.44 $km^2$ for anchorage, and changed from 38.52 $km^2$ to 15.47 $km^2$ for shipping channel [83]. | Almost 400 ha of land will be reclaimed for the construction of Tuas Megaport in 2019 [84]. |
| | | Noise | The noise at the port entrance in the Haicang area was slightly high [44], but port noise initiatives are emphasized in Xiamen [84]. | Port noise initiatives have not been highlighted in Singapore [85]. |

<div align="center">

**Table A2.** *Cont.*

</div>

| Activities | Indicators | | A Description of Conditions of Xiamen International Shipping | A Description of Conditions of Other Cases |
|---|---|---|---|---|
| Rail or road transportation | Non-living environment | Air | The mean volatile organic compounds concentration in the transport area was higher than that in the port zone and residential zone [79]. Transportation contributed to 55% of reactive nitrogen emissions to the atmosphere in Xiamen [86]. The emission intensities of $NO_2$ and CO for road transportation by truck were respectively estimated at 2.45 g/t·km and 9.39 g/t·km, which were lower than the provisions of the Environmental Impact Assessment Guidelines for Highways in China [14]. | The hydrocarbon-like organic aerosol concentrations and the co-emissions of sodium with refractory black carbon-containing particles in Singapore were greatly affected by road transportation [87]. |

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
