# Peer review of "An Ecological Sustainability Assessment Approach for Strategic Decision Making in International Shipping"

_sustainability, doi:10.3390/su132011471_

Round 1
Reviewer 1 Report
The manuscript brings a new important aspect on international shipping. A model is developed and tested using a case study. The article is coherent and well written.
Author Response
Thank you for your work. We have checked each comment, and some revisions were made in the manuscript (with changes marked). Responses to the comments were listed below:
Comments: The manuscript brings a new important aspect on international shipping. A model is developed and tested using a case study. The article is coherent and well written. 

Response: Thanks for the reviewer’s appreciation of our manuscript.
Comments: English language and style are fine/minor spell check required.
Response: We appreciate the reviewer’ s comment. We checked the English language and sent the manuscript to MDPI Webshop for English language editing. The English changes have been marked red in the manuscript, enclosing the editing service certificate. Thanks again for your comments.

Reviewer 2 Report
Thank you doe submitting your paper, “An ecological sustainability assessment approach for strategic 2 decision making in international shipping,”. Although your topic is interesting, your paper lacks on several fronts:
- When it comes to “ecological sustainability assessment,” authors missed even mentioning the most relevant methodology, “System Dynamic Approach” (Forrester, 1961; Sterman, 2000).
- In essence. “ecological sustainability” is a dynamic task and its evaluation or assessment with a non-dynamic methodology hardly is of any help.
- Also, ignoring the casualty in such an assessment is simply to show a too simplistic view of a complex, and dynamic task. Ecological sustainability owes to a log-term view and understanding of the casual nature of interactions among several relevant variables of shipping businesss (e.g., demand, size of containers, material used, after-use treatment, mode of transportation, regulations, etc., all require the understanding of their causal impacts on each other and then impacting the ecological sustainability.
- Also, the authors did not follow any scienetic method for their search, bor they cited any, inclusion/exclusions criteria for the review that forms the basis of this article.
I hope these comments will help you improve the content band quality of your paper for any future submission.
Author Response
Thank you for your work. We have checked each comment, and some revisions were made in the manuscript (with changes marked). Responses to the comments were listed below:
Comments: Moderate English changes required. 

Response: We appreciate the reviewer’s comment. We checked the English language and sent the manuscript to MDPI Webshop for English language editing. The English changes have been marked red in the manuscript, enclosing the editing service certificate. Thanks again for your comments.
Comments: Thank you doe submitting your paper, “An ecological sustainability assessment approach for strategic decision making in international shipping,”. Although your topic is interesting, your paper lacks on several fronts: When it comes to “ecological sustainability assessment,” authors missed even mentioning the most relevant methodology, “System Dynamic Approach” (Forrester, 1961; Sterman, 2000). In essence. “ecological sustainability” is a dynamic task and its evaluation or assessment with a non-dynamic methodology hardly is of any help. Also, ignoring the casualty in such an assessment is simply to show a too simplistic view of a complex, and dynamic task. Ecological sustainability owes to a log-term view and understanding of the casual nature of interactions among several relevant variables of shipping business (e.g., demand, size of containers, material used, after-use treatment, mode of transportation, regulations, etc., all require the understanding of their causal impacts on each other and then impacting the ecological sustainability.
Response: Thanks for the reviewer’s appreciation and comments. We agree that ecological sustainability assessment is in dynamic causal nature and a system dynamic approach has a great potential to assess the interactions between international shipping and ecosystems. It would be very interesting to explore the possible contribution of dynamic approaches to support ecological sustainability. However, as limitations currently exist in the assessment process related to international shipping such as biases of data acquisition and restrictions of causality identification confined by time and funds, our study aims to uncover a way in which a strategic decision-making system can be facilitated by combing through a broad scope of information as well as expert knowledge. It initially provides an ecology-based assessment approach for understanding the sustainability status in international shipping. Nevertheless, diving deep into dynamic relationships among several relevant variables of shipping can definitely contribute to sustainability assessment in international shipping, and we are going to further it by using a causal diagram or system archetype to provide a better basis for decision-making. These are reflected in future studies. Please refer to the revised paragraphs (Lines 273-288 and Lines 316-326).
Comments: Also, the authors did not follow any scientific method for their search, bor they cited any, inclusion/exclusions criteria for the review that forms the basis of this article. I hope these comments will help you improve the content band quality of your paper for any future submission.
Response: Thanks for the reviewer’s comments. The authors conducted a broad literature search in the Web of Science and combined a content analysis of abstracts to identify and compare the existing sustainability assessment methods (Lines 60-77 and Section 2). As a result, an MDDM approach that combines a top-down strategic ecological sustainability focus and a bottom-up broader data collection can better fit the strategic decision-making in the absence of sufficient data as well as under the current restrictions of causal identification. We picked up some representative articles as references. For all that, the literature search and analysis really spent us a considerable amount of time and we’d better follow a scientific method to refine the results of the literature search in future. Thanks for your suggestion. We really appreciate your help.

Round 2
Reviewer 2 Report
Thank you for your revised submission.
When you say, “However, as limitations currently exist in the assessment process related to international shipping such as biases of data acquisition and restrictions of causality identification confined by time and funds, our study aims to uncover a way in which a strategic decision-making system can be facilitated by combing through a broad scope of information as well as expert knowledge.”, this is the precisely the reason that in the current form your paper hardly makes any significant contribution to the knowledge. Yes, addressing these issues will certainly improve the quality of your paper.
Author Response
Many thanks for the reviewer’s meticulous work. We have checked each comment, and some revisions were made in the manuscript (with changes marked yellow). Responses to the comments were listed below:
Comments: English language and style are fine/minor spell check required. 

Response: Thanks for your comments. We checked the English language and made some revisions in Line 25, 45, 53, 98, 113, 210, 281, and 295.
Comments: When you say, “However, as limitations currently exist in the assessment process related to international shipping such as biases of data acquisition and restrictions of causality identification confined by time and funds, our study aims to uncover a way in which a strategic decision-making system can be facilitated by combing through a broad scope of information as well as expert knowledge.”, this is the precisely the reason that in the current form your paper hardly makes any significant contribution to the knowledge. Yes, addressing these issues will certainly improve the quality of your paper.
Response: We appreciate your comments. This manuscript aims to present an ecological sustainability assessment approach as a complement to the existing sustainability assessment methodology to serve the strategic decision-making associated with international shipping under current situations of literature. We highlighted the contributions of this study in Lines 20-22, 276-279, and 306-308. Thanks again for your comments.